# Refining Heuristic-Based Bitcoin Address Clustering with Graph Neural Networks

## Abstract

Bitcoin's pseudonymous nature makes it challenging to analyze user-level activity, since a single user may control multiple identifiers (addresses). Existing heuristic-based methods attempt to identify addresses belonging to the same user, but they often produce flat cluster assignments with limited modularity and are prone to errors such as merging different users together. In this work, we propose a method for refining heuristic-obtained clusters by grounding our clustering on contrastive embeddings yielded by graph neural networks. Our contribution is threefold: (i) we release a publicly available dataset of Bitcoin transaction graphs containing a substantial number of clusters; (ii) we propose a methodology for learning address embeddings consistent with heuristics, and back it up with solid theoretical foundations and empirical results; (iii) through hierarchical clustering, we allow a finer analysis of heuristic clusters and provide a quantitative criterion for flagging suspicious merges.

## 1 Introduction

Bitcoin (Nakamoto, 2009) is the first and most widely adopted cryptocurrency, designed as a decentralized payment system without reliance on a central authority. Its operation is enabled by a peer-to-peer network that collectively maintains a shared, immutable record of transactions (Antonopoulos, 2017a). This record, known as the blockchain, provides transparency and auditability while preserving a certain level of pseudonymity for its users; it is organized as a chronological sequence of blocks, each batching the transactions that happened during a certain time interval.

**Bitcoin Address Clustering.**    Bitcoin transactions are pseudonymous in nature, as users are identified by random pseudonyms called addresses (Antonopoulos, 2017b). A single user can reuse an address or generate new ones at any time; it is therefore common for a user to control many different addresses. Since addresses are generated randomly, there is no direct way to associate multiple addresses with the same user. While analyzing transaction at the address level can be informative, a user-level analysis provides greater insights. The task of *addresses clustering* consists in grouping together addresses that belong to the same user (without necessarily identifying said user).

**Graph Construction from Transactions.**    Graph-based representations are particularly well suited for visualizing and analyzing blockchain data. Two primary types of graphs are commonly employed: those where nodes represent transactions and edges represent the moving bitcoin amounts (Weber et al., 2019), and those where nodes represent users and edges represent transactions (Bellei et al., 2024; Schnoering & Vazirgiannis, 2025). In this paper, we focus on the latter, as it offers a more intuitive representation. Constructing a user-level graph from a set of transactions $\mathcal{T}$ typically involves the following steps (Schnoering & Vazirgiannis, 2025; Bellei et al., 2024; Meiklejohn et al., 2013; Harrigan & Fretter, 2016):

1. extracting the addresses involved in the transactions $\mathcal{T}$;
2. clustering the addresses into users using a heuristic $\mathcal{H}$ (or a combination thereof) applied to $\mathcal{T}$, potentially augmented with external information;
3. creating directed edges with associated features between users, derived from the $\mathcal{T}$;
4. generating node features by aggregating information from edges;
5. incorporating external information (off-chain) into both node and edge features.

**Hierarchical Clustering.** Hierarchical clustering constructs a hierarchy of nested clusters over a set of points $V$ endowed with a dissimilarity function $d$ (Heller & Ghahramani, 2005). In the agglomerative variant, each node initially forms its own cluster. At each step, two clusters $A, B \subset V$ are merged according to a linkage rule based on $d$. After the final step, all nodes are merged into a single cluster. This hierarchy is naturally represented by a rooted binary tree, or *dendrogram*, where leaves correspond to individual nodes, internal nodes represent successive merges, and node height indicates the merge distance. An example of dendrogram is illustrated in Figure 1.

**Graph Neural Networks (GNNs).** GNNs extend neural architectures to graph-structured data by propagating and transforming node features along edges. At each layer, a node updates its representation by aggregating information from its neighbors, allowing the model to capture both local connectivity and node attributes. By stacking multiple layers, GNNs learn embeddings that encode multi-hop structural context and can be used for tasks such as node classification, link prediction, and graph-level inference (Kipf, 2016; Hamilton et al., 2017; Veličković et al., 2017).

**Contributions.** The main contributions of this paper are threefold:

1. We publicly release a dataset of large-scale Bitcoin transaction graphs with a substantial number of clusters, enabling the training and evaluation of clustering algorithms at scale.
2. We propose a methodology for learning address embeddings consistent with traditional blockchain heuristics, supported by theoretical guarantees and empirical validation.
3. We show how these learned representations can refine heuristic-based clustering by detecting and correcting cluster collapses and by providing a hierarchical clustering that improves intelligibility and visualization.

## 2 RELATED WORKS

**Heuristics-Based Clustering.** To achieve address clustering, a variety of human-made, rule-based heuristics have been proposed (Schnoering et al., 2024), often based on behavioral patterns and human biases. The most prominent is the *common-input heuristic*, which assumes that all addresses providing inputs to the same transaction are controlled by a single entity. Clustering heuristics play a crucial role in Bitcoin analysis by approximating user-level structures from pseudonymous transaction data. They allow researchers and investigators to reduce complexity, uncover patterns of address ownership, and make sense of large-scale transaction graphs. Beyond their methodological value, such heuristics have become essential tools in several domains: in forensic contexts (Meiklejohn et al., 2013; Foley et al., 2019); in compliance and anti–money-laundering efforts (Möser et al., 2013; Yang et al., 2023), and in privacy research (Androulaki et al., 2013).

**Other Methods for Address Clustering.** Aside from heuristic clustering, other methods have been used on bitcoin transaction networks to similar tasks. Machine-learning based methods tend to focus more on the orthogonal task of address classification (Toyoda et al., 2018; Lin et al., 2019; Garin & Gisin, 2023; Sie et al., 2025; Jia et al., 2018; Lee et al., 2020), which consists in identifying the usage of addresses (e.g. scams, marketplaces). Some of those approaches (Kang et al., 2020) use heuristic clustering as a first step before training a classifier. More recently, approaches leverage GNNs to obtain powerful representation of transaction graphs for downstream tasks (Zhao et al., 2025; Zhang et al., 2025; Huang et al., 2022).

**Enhancing Clustering Heuristics with GNNs.** Despite their usefulness, heuristic methods have notable limitations. They yield only *flat* cluster assignments—single-level groupings in which addresses are either linked or not—making large clusters difficult to interpret. Some heuristics also merge addresses based on a single transaction, which can erroneously combine unrelated users and cause cluster collapse (Androulaki et al., 2013; Harrigan & Fretter, 2016). Only a few studies attempt to refine or correct the traditional heuristics. Möser & Narayanan (2022) use a random forest to estimate the likelihood that a heuristic-based merge is valid and block merges with low confidence, thereby mitigating cluster collapse. Similarly, Ermilov et al. (2017) uses off-chain information as votes for separating clusters.

Our method differs in key ways. Instead of assigning confidence scores to individual merges, we learn address embeddings that capture the global transaction structure while staying consistent with

heuristic clusters. Agglomerative hierarchical clustering on these embeddings yields a dendrogram that reveals nested substructures and provides a principled criterion for detecting suspicious merges, producing both a refined flat clustering and a multi-resolution view of the address graph.

## 3 METHODOLOGY

### 3.1 METHODOLOGY OVERVIEW

We present a method to learn address embeddings consistent with standard heuristics, mapping nodes from the same cluster close together and pushing nodes from different clusters apart. These embeddings are then used to build dendrograms whose hierarchical structure reveals discrepancies in the heuristic partitions—most notably cases of cluster collapse—and to propose corresponding corrections. Throughout the paper, let $G = (V, E)$ denote the graph, where $V$ is the set of nodes (Bitcoin addresses) and $E$ the set of edges (value transfers). We write $\mathcal{C} = \{C_1, \ldots, C_k\}$ for a partition of $V$ (e.g., obtained via heuristics), with $k$ the number of clusters.

**Rationale for the Two-Stage Methodology.** Our approach is in line with a broad body of prior work and offers a key practical advantage: it naturally accommodates dynamic graphs with continuously arriving addresses and transactions, closely reflecting real-world blockchain conditions. In contrast, most end-to-end GNN pooling methods (Ying et al., 2018; Bianchi et al., 2020) construct a fixed hierarchy of merged nodes whose depth and cluster sizes are predetermined by the network architecture. Such constraints hinder adaptation to a continually growing transaction graph and reduce the interpretability of the resulting merges. Other pooling approaches (Lee et al., 2019) merely score and retain important nodes without producing a true hierarchical clustering, offering saliency rather than an interpretable dendrogram of successive merges.

### 3.2 DATA ACQUISITION AND GRAPH CONSTRUCTION

We construct our graphs using the pipeline of Schnoering & Vazirgiannis (2025)[1]. The procedure follows the steps outlined in the introduction—parsing the blockchain, extracting transactions, and forming entity-to-entity links—but, unlike the original work, we do not pre-cluster addresses into user entities. The resulting network is a directed graph with nodes as addresses. User clusters serving as ground truth for supervised learning are obtained with the same set of address-clustering heuristics as in Schnoering et al. (2024), also implemented in the above GitHub repository. Constructing a graph from the entire history would yield billions of nodes and edges, rendering most algorithms intractable. We therefore sample a subset of transactions from a contiguous block interval to build the graph; the sampling strategy is described in the Appendix A.1.1. For complete implementation details, we refer readers to the original paper and accompanying code. The raw blockchain data for graph construction and clustering were obtained by running Bitcoin Core[2].

### 3.3 LEARNING NODE EMBEDDINGS WITH GNNS AND CONTRASTIVE LOSS

We train a GNN $g$ to produce node embeddings consistent with the clustering $\mathcal{C}$: nodes within the same cluster (user) should have similar embeddings, whereas embeddings of nodes from different clusters should be dissimilar. To enforce this, we adopt the contrastive InfoNCE loss (Oord et al., 2018; Chen et al., 2020)

$$\mathcal{L} = \mathbb{E}_{\mathbb{P}_\alpha}\left[ -\log \frac{\exp\big(g(X)\cdot g(X^+)/\tau\big)}{\exp\big(g(X)\cdot g(X^+)/\tau\big) + \sum_{i=1}^{p}\exp\big(g(X)\cdot g(X_i^-)/\tau\big)} \right], \tag{1}$$

where $\mathbb{P}_\alpha$ is the sampling distribution over anchor nodes, $\tau$ is a temperature hyperparameter, and $p$ is the number of negative samples. For each anchor $X \in V$, the positive sample $X^+$ is drawn from the same cluster, while the negatives $\{X_i^-\}_{i=1}^{p}$ come from different clusters. Clusters are drawn from a mixture of uniform and size-proportional sampling controlled by $\alpha$, and nodes are then sampled uniformly within each chosen cluster. Full details of this sampling scheme are provided in Appendix.

---

[1] https://github.com/hugoschnoering2/BTCGraphConstruction
[2] https://bitcoin.org/en/bitcoin-core

Although the formula omits explicit normalization, we normalize embeddings in practice so that the dot product computes cosine similarity.

### 3.4 Detecting and Correcting Cluster Collapse

We perform agglomerative hierarchical clustering on the embeddings using cosine distance, consistent with the contrastive loss. Starting from the coarse partition $\mathcal{C}$, we cluster each $C_i$ independently, building a dendrogram that records the merge distances within every initial community.

Given a threshold $\lambda > 0$, we define a *collapse* as any merge whose cosine distance exceeds $\lambda$. This provides a principled way to flag suspicious merges—likely combining addresses from different users—and highlights potential failures of the original flat clustering. To correct such collapses, we split the affected clusters into their hierarchical subcomponents, yielding a refined partition that better reflects the true user structure.

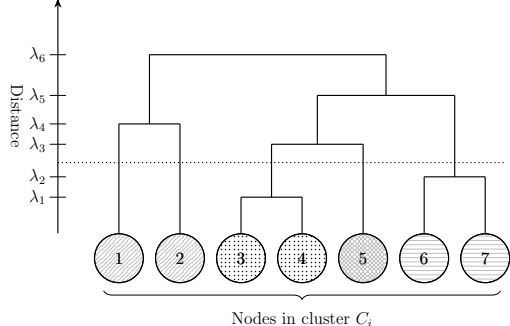

Mathematically, each dendrogram induces an ultrametric $d_u$ on the node set $V$, where $d_u(x, y)$ is the height of the lowest common ancestor of $x$ and $y$. Two nodes $x$ and $y$ are grouped together if they belong to the same initial cluster $C_i$ and satisfy $d_u(x, y) < \lambda$. This refinement process is illustrated in Figure 1.

Figure 1: Example of a refinement. The dotted line represents the cut. Sub-clusters are distinguished by node fill patterns. Merges above the threshold are treated as collapses.

A practical variant of this approach uses heuristic-generated clusters as the initial partition, motivated by the observation that such heuristics often merge distinct communities (i.e., distinct Bitcoin users).

## 4 Theoretical Foundations of the Methodology

We show that node embeddings learned by GNNs naturally separate nodes according to cluster membership in a hierarchical dendrogram, under appropriate conditions. Let $d$ be the working distance on $V$, and build a dendrogram from $d$ using single, average, or complete linkage. Assume the ground-truth clusters are well $d$-separated: there exist constants $0 < r < s$ such that $d(x, y) \leq r < s \leq d(x, z)$ for all $x, y \in C_\ell$ and every $z \in C_m$ with $\ell \neq m$. It then follows that any horizontal cut of this dendrogram at a threshold $\lambda \in (r, s)$ exactly recovers $C$; the resulting flat clustering coincides with the ground truth. Although these conditions are stronger than typically encountered in practice, they provide a clean theoretical framework for the analysis that follows and already motivate the use of a contrastive loss. The proofs of these results are provided in Appendix D.

**Notation.** Let $L$ be the Laplacian of $G$ with eigenvalues $\lambda_1 \leq \lambda_2 \leq \cdots \leq \lambda_n$ and associated orthonormal eigenvectors $u_1, \ldots, u_n$, which form an orthonormal basis of $\mathbb{R}^n$. Let $U \in \mathbb{R}^{n \times n}$ be the matrix whose columns are these eigenvectors. The spectral decomposition of $L$ is $L = UDU^\top$, where $D = \mathrm{diag}(\lambda_1, \ldots, \lambda_n)$ is the diagonal matrix of eigenvalues. Let $U_k \in \mathbb{R}^{n \times k}$ be the matrix formed by the first $k$ eigenvectors. For a node $i \in V$, its spectral embedding is $e_i^s = (u_{i,1}, u_{i,2}, \ldots, u_{i,k}) \in \mathbb{R}^k$, where $k$ is the number of clusters in the partition $C$. We write $\|x\|_2$ for the Euclidean norm of a vector $x$. For any matrix $A$, $A^\top$ denotes its transpose, $\sigma_{\min}(A)$ the smallest singular value of $A$, and $\|A\|_{\mathrm{op}}$ for the operator norm of $A$ induced by $\|\cdot\|_2$.

### 4.1 Results

Building on the perfect–cut criterion above, our goal is to derive a separability condition on the problem data that guarantees a dendrogram built from GNN embeddings admits such a perfect cut. Both results in this section assume that the working distance is Euclidean. The arguments, however, remain valid for cosine distance provided that the GNN embeddings lie on a common sphere. As a

first step, Lemma 1 establishes an analogous condition for spectral embeddings. This intermediate result is natural because GNNs typically act as low-pass spectral filters (Nt & Maehara, 2019), so their embeddings concentrate in the subspace spanned by the Laplacian eigenvectors with the smallest eigenvalues, i.e., the classical spectral embeddings (Von Luxburg, 2007). The result involves the spectral distance between the Laplacian $L$ and the Laplacian $L^\circ$ of an *ideal cluster graph*, where two nodes are connected if and only if they belong to the same cluster. This ideal graph represents a perfectly homophilic scenario in which edges exist only within clusters. The appearance of this quantity is motivated by empirical observations on data, where addresses controlled by the same user tend to form connected subgraphs.

**Lemma 1.** *The spectral embeddings are cluster–separable whenever*

$$M := 4\sqrt{2k}\left(1 - \tfrac{1}{S_{\max}}\right)\|L - L^\circ\|_{\mathrm{op}} < \frac{1}{\sqrt{2S_{\max}}}.$$

*where $S_{\max}$ is the size of the largest cluster, and $L^\circ$ the Laplacian of the ideal cluster graph.*

The proof in Appendix D.1 relies on a version of the Davies–Kahan theorem from matrix perturbation theory. The separability condition is satisfied whenever the graph Laplacian $L$ is sufficiently close to the ideal block–diagonal Laplacian.

We assume that the node embeddings $H$ produced by the GNN can be written as $H = p(L)\,XW$, where $p$ is a polynomial, $X$ the matrix of initial node features, and $W$ the learned weight matrix (as in the linearized GCN (Kipf, 2016), for example). Using the spectral decomposition $L = UDU^\top$, this becomes

$$H = U\,\tilde{D}\,U^\top XW,$$

where $\tilde{D} = \mathrm{diag}(p(\lambda_1), \ldots, p(\lambda_n))$. The polynomial $p$ acts as a *spectral filter*, selectively amplifying or attenuating the eigencomponents of $L$ according to their eigenvalues. In the special case of an ideal low-pass filter, $p(\lambda_i) = \mathbf{1}_{\{i \leq k\}}$, so the embeddings lie entirely in the subspace spanned by the first $k$ eigenvectors. To measure how well a GNN approximates this ideal filter, we define $\alpha = \max_{i \leq k}|p(\lambda_i)|$, $\beta = \max_{i > k}|p(\lambda_i)|$, and $\gamma = \min_{i \leq k}|p(\lambda_i)|$. Theorem 2 transfers this spectral result to the learned GNN embeddings, yielding an equivalent separability condition for the perfect cut—a result that, to our knowledge, is novel.

**Theorem 2.** *The GNN embeddings are cluster–separable whenever*

$$\|XW\|_{\mathrm{op}}(\beta + \alpha M) \; < \; \gamma\,\sigma_{\min}(U_k^\top XW)\left(\sqrt{2/S_{\max}} - M\right).$$

The embeddings learned by the GNN inherit the geometric separability of the spectral embeddings, up to perturbations controlled by the low-pass approximation quality of $p$ and by the alignment of the feature matrix $XW$ with the leading eigenspace. Because the left-hand side of the inequality is positive, the separability condition holds only if three requirements are met: (i) $\gamma > 0$, so the GNN retains all eigencomponents of the informative subspace; (ii) $\sigma_{\min}(U_k^\top XW) > 0$, ensuring that the transformed features are not orthogonal to this subspace; and (iii) $M \leq \sqrt{2/S_{\max}}$, meaning the observed graph is sufficiently close to the ideal block-diagonal Laplacian so that spectral embeddings themselves already separate the clusters.

## 4.2 RELATED WORKS

Spectral embeddings have long been central to graph clustering (Von Luxburg, 2007). Most theoretical analyses relate these embeddings to the *optimal* solutions of node-partitioning problems, including RatioCut minimization (Von Luxburg, 2007), $k$-way partitioning (Peng et al., 2015), and maximum-margin clustering (Hofmeyr, 2020). The guarantees in these works require the reference clustering to coincide with the optimal solution of the respective problem. Our approach makes no such assumption. We instead study graphs that are small perturbations of an *ideal cluster graph* whose connected components match the ground-truth clusters, and we apply matrix perturbation theory to obtain our guarantees. This technique was also used by Ng et al. (2001) to bound intra-cluster variance. In contrast, we establish *pairwise* bounds—both within and across clusters—yielding separability conditions that ensure a perfect cut.

## 5 Experimental Setup

All experiments were performed on a Mac M3 Max equipped with 36 GB of RAM, using only CPU computation and no GPU acceleration.

We use the pipeline described in Section 3 to generate graphs from Bitcoin transactions. To avoid information leakage, transaction sets are sampled from non-overlapping block intervals so that no transaction appears in more than one graph. In total, we construct three graphs for training, one for validation, and one for testing. The main characteristics of these graphs are provided in Appendix A, and all datasets, including the graphs used in the experiments with ground truth labels in Section 6.3, are publicly available at ⋆⋆⋆ under the CC BY 4.0 license.

Before being fed to the GNNs, features undergo the normalization and log-scaling procedure detailed in Appendix B.2. This step ensures consistent feature distributions across the different graphs.

### 5.1 Training

**Setup.** We train two-layer GNNs to minimize the contrastive loss of Equation equation 1, monitoring progress by evaluating the same loss on a validation graph. We experiment with three popular architectures: Graph Convolutional Network (GCN) (Kipf, 2016), GraphSAGE (Hamilton et al., 2017), and Graph Attention Network (GAT) (Veličković et al., 2017). Optimization uses Adam (Kingma & Ba, 2014) with a learning rate halved when the validation loss does not improve for 20 consecutive epochs. Because we have three training graphs, we cycle through them every 15 epochs to promote generalization. To accelerate training, we adopt neighborhood sampling (Hamilton et al., 2017), drawing 15 neighbors for the first GNN layer and 5 for the second. All experiments rely on the `PyTorch Geometric` implementations of GNN models, the Adam optimizer, learning-rate scheduler, and neighbor sampling. The code used in this study is publicly available at ⋆⋆⋆. Unless otherwise specified, all hyperparameters are listed in Table 6 of Appendix B.3.

**Model Variations.** We evaluate three main variations of the base model. (1) Because the constructed graphs are directed, we optionally symmetrize them before input to the GNN. (2) Since edges carry attributes, we can include or ignore these edge features whenever the architecture supports them. (3) We optionally add a structural positional encoding to enhance locality. GNN message passing tends to make nodes with similar neighborhoods appear similar—even when they are far apart (Xu et al., 2019)—which can spuriously cluster structurally alike but unrelated nodes. Yet Bitcoin addresses belonging to the same user are usually close in the graph, as they often participate in the same transactions. To exploit this property, we follow the position-aware GNN framework (You et al., 2019): we select the highest-degree nodes as landmarks and represent each node by its vector of shortest-path distances to these landmarks. These distances are converted to similarities via $x \mapsto (1 + x)^{-1}$ and normalized dimension-wise. The resulting distance-based vector is then concatenated with the original node feature vector before message passing.

### 5.2 Evaluation

We evaluate our method by its ability to recover both hierarchical and flat clusterings consistent with the ground truth. For the hierarchical step, we apply agglomerative clustering with cosine distance on the GNN embeddings using average linkage. Because the graphs are large and computing the full pairwise distance matrix is impractical, we first obtain a coarse partition with the Leiden algorithm (Traag et al., 2019), limiting the maximum community size to 65 000 nodes to control memory usage, following the strategy described in Section 3.4.

**Metrics.** We score the resulting dendrograms with *dendrogram purity* (Heller & Ghahramani, 2005), which ranges from 0 to 1 and measures how well nodes from the same ground-truth cluster merge together. Flat clusterings are obtained by cutting each dendrogram at a threshold $\lambda$ (Figure 1), selected by grid search to maximize the silhouette score (Rousseeuw, 1987), a standard criterion for choosing the cut level in hierarchical clustering. We then compare the flat partition to the ground truth using Normalized Mutual Information (NMI) and Adjusted Rand Index (ARI) (Vinh et al., 2009): NMI captures global agreement and is robust to cluster-size imbalance, while ARI emphasizes local consistency but is more sensitive to class imbalance. Additional implementation details

| Model | Sym. | Edge feat. | # LM | DP | NMI | ARI |
|---|---|---|---|---|---|---|
| Louvain | ✓ | na | na | na | 0.642 ($\pm$0.000) | 0.289 ($\pm$0.000) |
| Leiden | ✓ | na | na | na | 0.665 ($\pm$0.000) | 0.311 ($\pm$0.000) |
| Random GAT | ✓ | × | 0 | 0.691 ($\pm$0.004) | 0.759 ($\pm$0.011) | 0.586 ($\pm$0.009) |
| GAE | ✓ | × | 0 | 0.741 ($\pm$0.006) | 0.746 ($\pm$0.006) | 0.325 ($\pm$0.213) |
| DIG | ✓ | × | 0 | 0.684 ($\pm$0.004) | 0.755 ($\pm$0.008) | 0.607 ($\pm$0.041) |
| GAT | × | × | 0 | 0.649 ($\pm$0.011) | 0.714 ($\pm$0.010) | 0.320 ($\pm$0.043) |
| | × | × | 64 | 0.689 ($\pm$0.003) | 0.745 ($\pm$0.004) | 0.611 ($\pm$0.025) |
| | × | × | 128 | 0.693 ($\pm$0.002) | 0.743 ($\pm$0.004) | 0.590 ($\pm$0.004) |
| | × | × | 256 | 0.685 ($\pm$0.002) | 0.741 ($\pm$0.003) | 0.589 ($\pm$0.002) |
| GAT | × | ✓ | 0 | 0.642 ($\pm$0.011) | 0.705 ($\pm$0.012) | 0.333 ($\pm$0.046) |
| | × | ✓ | 64 | 0.688 ($\pm$0.002) | 0.743 ($\pm$0.003) | 0.593 ($\pm$0.003) |
| | × | ✓ | 128 | 0.692 ($\pm$0.004) | 0.743 ($\pm$0.004) | 0.596 ($\pm$0.006) |
| | × | ✓ | 256 | 0.686 ($\pm$0.004) | 0.739 ($\pm$0.004) | 0.587 ($\pm$0.003) |
| GAT | ✓ | × | 0 | 0.783 ($\pm$0.004) | 0.775 ($\pm$0.008) | 0.702 ($\pm$0.033)[*] |
| | ✓ | × | 64 | 0.796 ($\pm$0.003)[**] | 0.770 ($\pm$0.002) | 0.707 ($\pm$0.012)[**] |
| | ✓ | × | 128 | 0.793 ($\pm$0.002) | 0.770 ($\pm$0.008) | 0.672 ($\pm$0.029) |
| | ✓ | × | 256 | 0.792 ($\pm$0.002) | 0.771 ($\pm$0.003) | 0.665 ($\pm$0.029) |
| GAT | ✓ | ✓ | 0 | 0.778 ($\pm$0.005) | 0.782 ($\pm$0.012)[*] | 0.677 ($\pm$0.031) |
| | ✓ | ✓ | 64 | 0.794 ($\pm$0.003)[*] | 0.765 ($\pm$0.005) | 0.661 ($\pm$0.052) |
| | ✓ | ✓ | 128 | 0.789 ($\pm$0.004) | 0.771 ($\pm$0.004) | 0.691 ($\pm$0.043) |
| | ✓ | ✓ | 256 | 0.787 ($\pm$0.005) | 0.765 ($\pm$0.004) | 0.639 ($\pm$0.053) |
| GCN | ✓ | × | 0 | 0.724 ($\pm$0.003) | 0.767 ($\pm$0.002) | 0.592 ($\pm$0.006) |
| GraphSage | ✓ | × | 0 | 0.768 ($\pm$0.004) | 0.791 ($\pm$0.002)[**] | 0.622 ($\pm$0.016) |

Table 1: Performance across different variations: graph symmetrization (Sym.), edge features (Edge feat.), number of landmarks (# LM), with evaluation metrics NMI, ARI, and dendrogram purity (DP), non applicable (na). The best score for each metric is marked with [**], the second-best with [*] and the performance of the model with all default parameters is underlined.

on how these metrics are computed, as well as their formal definitions, are provided in Appendix B.4. For all metrics we evaluate only nodes with degree $\geq 2$, excluding peripheral addresses that often lack sufficient transactional context for reliable user clustering and can artificially inflate cluster counts, making global metrics less informative.

**Baselines.** To highlight the added value of the contrastive loss, we compare our model to three unsupervised baselines: (i) an untrained GAT, (ii) a GAT trained as a non-probabilistic Graph Auto-Encoder (GAE) (Kipf & Welling, 2016), and (iii) a GAT trained with Deep Graph Infomax (DGI) (Veličković et al., 2018), which maximizes mutual information between local and global representations. All baselines produce node embeddings that are clustered exactly as in our contrastive pipeline. Implementation details are provided in Appendix B.5.

# 6 RESULTS

## 6.1 ABLATION STUDY

We report in Table 1 the performance results for different variations, including graph symmetrization, use of edge features, and the number of landmarks in the structural embedding. For each model variation, we averaged the results over five runs with different random seeds on the test graph. Additional experiments on the embedding dimension, the sampling parameter $\alpha$, and the number of negative anchors in the contrastive loss, as well as empirical evidence that our method approaches the conditions required by the theoretical results, are reported in Appendix C.

All baselines achieve ARI scores above zero—better than random—showing that graph topology alone conveys cluster information and supporting the homophily hypothesis. An untrained GAT already surpasses Louvain and Leiden, highlighting the strong signal in the input features. GAE yields higher dendrogram purity than the random GAT but lower NMI and ARI, consistent with its link-prediction loss, which encourages neighbors to share embeddings and can merge unrelated users. DGI matches the untrained GAT on dendrogram purity and NMI while achieving a stronger ARI, suggesting that its mutual-information objective promotes sharper local separation.

Results show that a GAT trained on a non-symmetrized graph generally performs worse on all metrics than an untrained GAT, highlighting the importance of reciprocal connections for capturing address relationships. Adding the structural positional encoding improves performance in the non-

symmetric setting, closing the gap with the untrained GAT. In contrast, incorporating edge features in the non-symmetric case offers no clear benefit and in fact slightly degrades performance.

All models trained on symmetrized graphs outperform the baselines on every metric. Using the structural positional encoding generally increases dendrogram purity—improving hierarchical clustering—while slightly reducing NMI and ARI, which measure flat clustering quality. This suggests that the silhouette score may be suboptimal for selecting the dendrogram cut. Among landmark-based encodings, the best results occur with 64 landmarks, followed by 128 and 256. The decline in performance as the number of landmarks increases—and the lack of gains from incorporating edge features despite their additional information—points to potential training instabilities or feature redundancy, highlighting the need for careful tuning.

For the alternative architectures, GCN and GraphSage, only the NMI score of GraphSage exceeds that of GAT under the same settings, supporting our choice of GAT as the primary architecture.

## 6.2 ILLUSTRATING CLUSTER REFINEMENT

We address potential cluster collapse using the procedure of Section 3. Starting from the heuristic clustering, we build a hierarchical clustering within each heuristic cluster and obtain a refined flat partition by cutting each dendrogram at the threshold $\lambda$ that maximizes the silhouette score. Figure 2 shows the resulting dendrogram for a representative cluster, with the selected cut level indicated. Its structure reveals the sequence of merges and highlights several late merges occurring above the optimal threshold. In particular, the final two subclusters merge at a cosine distance of 0.45, well above the chosen cut, indicating that they should remain separate. A few other merges also exceed the threshold, although most nodes merge below it into a single coherent group.

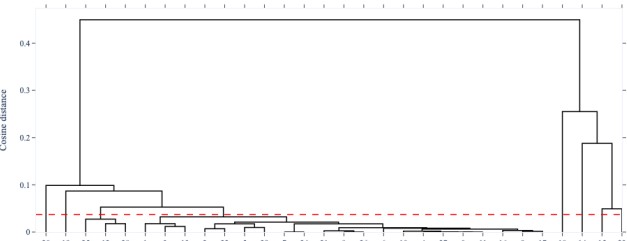

Figure 2: Dendrogram for a representative heuristic cluster. The dashed horizontal line indicates the cut level $\lambda$ selected to maximize the global silhouette score.

Figure 3 displays the minimal subgraph induced by the cluster and its neighbors. Cutting the dendrogram at the optimal threshold reveals coherent sub-groups, offering a clearer view of the cluster's internal organization. This approach naturally scales to much larger clusters, tens of thousands of nodes in our data and potentially millions in larger transaction sets, where direct graph visualization becomes impractical. Dendrograms provide a hierarchical, navigable representation that exposes meaningful substructures at multiple resolutions.

## 6.3 ADDITIONAL EXPERIMENTS WITH GROUND-TRUTH LABELS

In each experiment, we select transactions for which ground-truth labels indicate whether two addresses do or do not belong to the same cluster. For each transaction with labels, we extract a local transaction subgraph using a sampling procedure adapted from Section 3.2 (details in Appendix A.1.2), and construct the corresponding address-level graph. For each graph, we compute (1) the clustering from standard heuristics, (2) the clustering from our default GNN–HAC pipeline, and (3) a hybrid clustering where GNN embeddings refine the heuristic output, as described in Section 3.4. We evaluate three linkage criteria and three dendrogram-cutting strategies (Appendix **??**). Because some involved addresses are labeled, we can assess clustering quality with standard binary metrics: correct predictions group same-entity addresses or separate different ones, while errors correspond to incorrect merges or splits. To avoid overweighting transactions with many labels, we evaluate at most five randomly selected labeled pairs per graph.

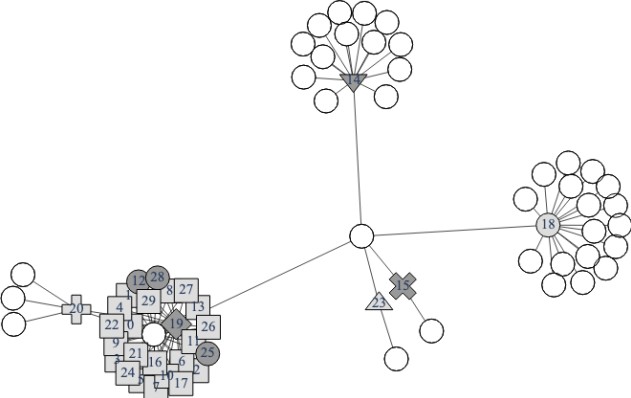

Figure 3: Minimal subgraph induced by the representative cluster and its immediate neighbors. Nodes belonging to the cluster are numbered, while external neighbors remain unnumbered. Cutting the dendrogram at the optimal threshold reveals distinct sub-groups, shown here with different gray shades and marker shapes.

| Model | Link. | Cut | TP(%) | FP(%) | FN(%) | TN(%) | bACC(%) | F1(%) |
|---|---|---|---|---|---|---|---|---|
| Heuristics | na | na | 42.9 | 22.5 | 16.0 | 18.6 | 59.1 | 59.2 |
| GNN-HAC | avg. | sil. | 46.0 | 20.9 | 12.9 | 20.1 | 63.7 | 63.8 |
| GNN-HAC | avg. | inc. | 58.9 | 41.1 | 0.0 | 0.0 | 50.0 | 37.1 |
| GNN-HAC | avg. | gap. | 52.1 | 28.7 | 6.8 | 12.4 | 59.4 | 57.9 |
| GNN-HAC | ward | sil. | 48.6 | 22.5 | 10.3 | 18.6 | 63.9 | 64.0 |
| GNN-HAC | com. | sil. | 38.8 | 16.1 | 20.1 | 25.0 | 63.4 | 63.1 |
| Hybrid | avg. | sil. | 40.7 | 11.7 | 18.2 | 29.4 | 70.3** | 69.7** |
| Hybrid | avg. | inc. | 42.9 | 22.5 | 16.0 | 18.6 | 59.1 | 59.2 |
| Hybrid | avg. | gap. | 42.4 | 18.5 | 16.5 | 22.6 | 63.5 | 63.5 |
| Hybrid | ward | sil. | 42.2 | 22.5 | 16.7 | 18.6 | 58.5 | 58.6 |
| Hybrid | com. | sil. | 37.6 | 10.6 | 21.3 | 30.5 | 69.0* | 67.9* |

Table 2: Clustering performance with ground-truth entity labels. Linkage (Link.) criteria: average linkage (avg.), Ward linkage (ward), and complete linkage (com.). Dendrogram cut methods: silhouette-based cut (sil.), inconsistency cut (inc.), and largest-gap cut (gap.). Metrics reported include true positives (TP), false positives (FP), false negatives (FN), true negatives (TN), along with balanced accuracy (bACC), defined as the mean of positive and negative recalls, and the macro-averaged F1 score (F1). The best score for each metric is marked with **, the second-best with *.

### 6.3.1 ENTITY LABELS

We use the dataset of roughly 100,000 addresses labeled with entity names from Schnoering & Vazirgiannis (2025). After excluding addresses linked to individuals, we sample 500 transactions between blocks 550,000 and 700,000 that involve at least two distinct labeled addresses. Addresses associated with the same entity should fall in the same cluster, while those linked to different entities should not. Results are reported in Table 2.

The average-linkage / silhouette-score setting yields substantial improvements over the heuristic baselines. The best results are obtained with the refinement pipeline, underscoring the importance of the hybrid approach: macro-F1 and accuracy increase by more than 10%, and false positives are reduced by half, effectively preventing cluster collapses. Complete linkage performs almost as well, with very similar results. In contrast, Ward linkage is more mixed: it appears particularly effective only when heuristic clusters are not further refined. The largest-gap criterion improves heuristic results only within the refinement step. Finally, the inconsistency criterion is not well suited to our task, as it leads to over-clustering, reflected by the high false-positive rate.

### 6.3.2 COINJOIN TRANSACTION LABELS

CoinJoin transactions involve many users and are specifically designed to defeat the common-input heuristic (Schnoering & Vazirgiannis, 2025). Based on analyses of open-source CoinJoin protocol implementations, Schnoering & Vazirgiannis (2023) proposed heuristics capable of detecting

| Model | Link. | Cut | TN(%) |
|---|---|---|---|
| Heuristics | na | na | 0. |
| GNN-HAC | avg. | sil. | 6.5 |
| GNN-HAC | ward | sil. | 11.5 |
| GNN-HAC | com. | sil. | 25.5 |
| Hybrid | avg. | sil. | 2.0 |
| Hybrid | ward | sil. | 0.3 |
| Hybrid | com. | sil. | 8.0 |

Table 3: Clustering performance with ground-truth CoinJoin labels. Results are shown for average linkage (avg.) combined with three dendrogram cut methods: silhouette-based cut (sil.), inconsistency cut (inc.), and largest-gap cut (gap.). The evaluation metric is the true-negative rate (TN).

most such transactions. In these transactions, all input addresses are expected to belong to different clusters. For each protocol examined in Schnoering & Vazirgiannis (2023), we randomly selected 100 transactions between blocks 550,000 and 700,000. As these transactions contain only negative cases, performance is measured solely through the true-negative rate. Classical heuristics without CoinJoin-aware protections, as noted in Schnoering et al. (2024), achieve a score of zero. Results for the different clustering methods are reported in Table 3.

Using the embeddings reduces the number of false positives arising from CoinJoin transactions. This effect is particularly strong in the setting without heuristic-cluster refinement, where complete linkage lowers false positives by roughly one quarter. In the refinement setting, however, the gain remains modest, either because the threshold selection is suboptimal or because the dendrogram structure does not sufficiently expose dubious merges. Robustness to CoinJoin transactions could also be improved by explicitly incorporating them as true negative examples in the contrastive loss.

## CONCLUSION AND LIMITATIONS

This work presents a principled framework for refining heuristic-based Bitcoin address clustering through contrastive GNN embeddings that remain consistent with standard heuristics while uncovering richer hierarchical structure. Starting from classical clustering rules, our method learns embeddings that separate users in latent space and applies agglomerative hierarchical clustering to reveal substructures and flag suspicious merges. Together, these elements provide a unified toolkit—data, theory, and methodology—for moving from flat heuristic clusters to interpretable, multi-resolution user graphs. A key limitation, however, is the limited amount of ground-truth labels available for evaluating user clusters at scale.

An important direction for future work is to adapt this procedure to a dynamic transaction graph that grows as new blocks and addresses appear, enabling online refinement of user clusters. A key challenge will be scalability. While node embeddings can be approximated by sampling subgraphs of manageable size, constructing the hierarchical structure is far less scalable, as illustrated in Appendix E. Nevertheless, this limitation can be partially mitigated by sampling very local subgraphs, either around specific transactions or within narrow temporal windows, which keeps graph sizes controlled while maintaining strong clustering performance. Future research should therefore focus on scalable hierarchical clustering techniques capable of handling continuously evolving blockchain graphs.

**LLM Usage** The research ideas, work and content presented in this paper was fully designed and made by the authors. LLMs were solely used in improving grammar and language a-posteriori, with no contribution besides reformulating for clarity purpose. In particular, no new idea or element was introduced through the use of an LLM.

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

# A DATASET

## A.1 TRANSACTIONS SAMPLING STRATEGY

### A.1.1 SAMPLING FROM COINBASE TRANSACTIONS

Constructing a graph from the full history of Bitcoin transactions would yield a network with several billions of nodes and edges, rendering most graph algorithms computationally infeasible. To obtain a manageable subgraph, we sample transactions occurring between two block indices $t_1 < t_2$.

A Bitcoin transaction transforms input value units into new output value units (TXOs), with unspent outputs known as UTXOs. Inputs originate from previous transactions, while outputs can be spent by future transactions. The only exception is the *coinbase transaction*—the first transaction in each block—which has no inputs and generates new currency units as a mining reward.

This structure naturally defines a directed acyclic graph (DAG): sources correspond to coinbase transactions; an edge exists from transaction $A$ to transaction $B$ whenever outputs from $A$ are consumed by $B$; sinks correspond to transactions whose no output has been spent. An example of such a transaction DAG is shown in Figure 4.

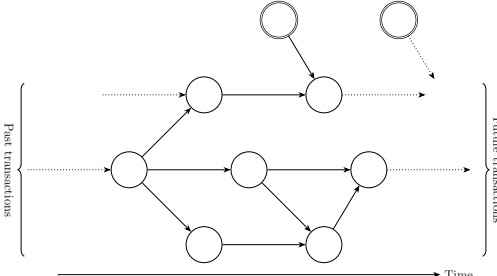

Figure 4: Bitcoin transaction graph. Circles represent transaction nodes, directed edges indicate the flow of bitcoin between transactions. Double–circled nodes denote coinbase transactions.

Our sampling procedure performs a breadth-first search (BFS) on this transaction DAG, initialized from a coinbase transaction chosen uniformly at random between blocks $t_1$ and $t_2$. Because block indices increase monotonically along transaction paths, all sampled transactions necessarily have indices greater than $t_1$, and exploration is truncated at block $t_2$, ensuring that every sampled transaction lies within the interval $[t_1, t_2]$. To further control the graph size, we cap the exploration depth at 15 and limit the number of transactions expanded at each BFS step to 5,000.

### A.1.2 SAMPLING FROM TRANSACTIONS WITH LABELS

In contrast to the directed exploration used for coinbase-based sampling, we perform a breadth-first search on the *undirected* transaction graph. This allows the procedure to explore not only future transactions consuming outputs of the seed, but also past transactions whose outputs were used as inputs to it.

To preserve locality, we use a maximum BFS depth of 3 and limit the number of expanded transactions per depth to 100. These constraints ensure that the sampled subgraph remains compact while capturing the relevant transactional context surrounding the labeled addresses. Aside from these modifications, the overall exploration logic follows the same structure as the coinbase-based sampling procedure described above in Appendix A.1.1.

## A.2 GRAPH CHARACTERISTICS

Table 4 summarizes the key statistics of the sampled Bitcoin transaction graphs used for training, validation, and testing.

| Split | Blk int. | Blk dates int. | #Tx | #Nodes | #Edges | #Clust. |
|---|---|---|---|---|---|---|
| Train (1) | [599k, 600k] | 12/10/19 - 19/10/19 | 57k | 643k | 5.3M | 105k |
| Train (2) | [624k, 625k] | 02/04/20 - 08/04/20 | 48k | 491k | 3.7M | 96k |
| Train (3) | [674k, 675k] | 10/03/21 - 17/03/21 | 44k | 923k | 15.1M | 164k |
| Valid | [649k, 650k] | 19/09/20 - 26/09/20 | 57k | 828k | 7.0M | 137k |
| Test | [699k, 700k] | 04/09/21 - 11/09/21 | 56k | 1071k | 4.2M | 141k |

Table 4: Dataset statistics. Abbreviations: Blk int. = block interval, #Tx = number of transactions sampled, #Nodes = number of nodes, #Edges = number of edges, #Clust. = number of clusters.

### A.3   NODE AND EDGE FEATURES

Table 5 lists the columns and their descriptions for each table (nodes, edges, and clusters) in the released graph dataset.

| Table | Column name | Description |
|---|---|---|
| Nodes | node_id | Identifier of the node |
| | degree_in | The number of incoming edges to the node |
| | degree_out | The number of outgoing edges from the node |
| | total_transaction_in | Total count of transfers received by the node |
| | total_transaction_out | Total count of transfers initiated by the node |
| | first_transaction_in | Block index of the first transfer received |
| | last_transaction_in | Block index of the last transfer received |
| | first_transaction_out | Block index of the first transfer sent |
| | last_transaction_out | Block index of the last transfer sent |
| | min_sent | Smallest value sent out in a single transaction |
| | max_sent | Largest value sent out in a single transaction |
| | total_sent | Cumulative value of all outgoing transfers |
| | min_received | Smallest value received in a single transaction |
| | max_received | Largest value received in a single transaction |
| | total_received | Cumulative value of all incoming transfers |
| Edges | a | Node id of the sender |
| | b | Node id of the recipient |
| | reveal | Block index of the first transaction |
| | last_seen | Block index of the last transaction |
| | total | Total number of transactions |
| | min_sent | Minimum sent in a single transaction |
| | max_sent | Maximum sent in a single transaction |
| | total_sent | Total sent in a single transaction |
| Clusters | node_id | Identifier of the node |
| | alias | Identifier of the cluster |

Table 5: Description of the columns of the different tables constituting the graph.

## B   TRAINING

### B.1   SAMPLING FUNCTION FOR CONTRASTIVE LEARNING

We assume a ground-truth clustering $\mathcal{C} = \{C_1, \ldots, C_k\}$ over the node set $V$. Let the latent variables $(Z, Z_1^-, \ldots, Z_p^-)$ denote the cluster labels of $(X, X_1^-, \ldots, X_p^-)$ under $\mathcal{C}$. The joint sampling distribution of equation 1 is

$$\mathbb{P}_\alpha(x, x^+, x_1^-, \ldots, x_p^-) = \sum_{z, z_1^-, \ldots, z_p^-} \mathbb{P}_\alpha(z)\, \mathbb{P}(x \mid z)\, \mathbb{P}(x^+ \mid z, x) \prod_{i=1}^{p} \mathbb{P}_\alpha(z_i^- \mid z)\, \mathbb{P}(x_i^- \mid z_i^-).$$

where $\mathbb{P}_\alpha(Z = z) = \alpha \frac{|C_z|}{|V|} + (1 - \alpha) \frac{1}{|\mathcal{C}|}$ is a mixture between size-proportional sampling ($\alpha = 1$), $\mathbb{P}(X = x \mid Z = z)$ is uniform over all nodes in cluster $C_z$, $\mathbb{P}(X^+ = x^+ \mid Z = z, X = x)$ is uniform over $C_z \setminus \{x\}$, $\mathbb{P}_\alpha(Z_i^- = z_i^- \mid Z = z)$ is uniform over all $z_i^- \neq z$ and $\mathbb{P}(X_i^- = x_i^- \mid Z_i^- = z_i^-)$ is uniform over all nodes in cluster $C_{z_i^-}$.

This scheme provides a principled sampling strategy for contrastive learning: positive pairs are always drawn from the same cluster as the anchor, while negatives come from different clusters. The parameter $\alpha$ balances diversity and representativeness by interpolating between uniform and size-proportional cluster sampling.

## B.2  FEATURES PREPROCESSING

Some input features encode amounts denominated in bitcoins (Table 5). Because the bitcoin price varies substantially across graph samples, we augment these features with their corresponding U.S.-dollar values, computed from the bitcoin price at each graph's starting date.

Feature preprocessing is performed independently for each graph. First, all features are log-transformed using $x \mapsto \log(1 + x)$ to reduce skewness. Next, we apply min–max normalization based on the empirical 5th and 95th percentiles of each feature, and missing values are imputed with zeros.

## B.3  HYPERPARAMETERS

Table 6 summarizes the model architecture, preprocessing options, and optimization settings used for training the GNNs.

|  | Hyperparameter | Value |
|---|---|---|
| Model | Number of attention heads | 4 |
|  | Size of hidden embeddings | 64 |
|  | Size of output embeddings | 128 |
|  | Number of layers | 2 |
|  | Activation function | Leaky ReLU |
|  | Dropout | 0.2 |
| Preprocessing | Symmetrize the input graph | True |
|  | Use edge features | False |
|  | Number of landmarks in the positional encoding | 0 |
| Optimizer | Initial learning rate | $2.5 \times 10^{-3}$ |
|  | Weight decay | $10^{-5}$ |
| Learning rate scheduler | Reduction factor | 0.5 |
|  | Patience | 20 |
| Gradient descent | Number of epochs | 250 |
|  | Num anchors per batch | 512 |
|  | Num negative samples per anchor ($p$) | 4 |
|  | Temperature ($\tau$) | 0.07 |
|  | Parameter of sampling function ($\alpha$) | 0.5 |

Table 6: Hyperparameters used in the training.

## B.4  EVALUATION METRICS

We evaluate hierarchical and flat clusterings using standard information–theoretic and pairwise similarity measures.

### B.4.1  HIERARCHICAL CLUSTERING

**Dendrogram Purity.**  Following Heller & Ghahramani (2005), let $T$ be a dendrogram with leaves $1, \ldots, n$ and class labels $c_1, \ldots, c_n$. To compute the purity of $T$:

1. Sample a leaf $\ell$ uniformly at random.
2. Sample another leaf $j$ uniformly at random among those with the same class label, $c_j = c_\ell$.
3. Let $S(\ell, j)$ be the smallest subtree of $T$ containing both $\ell$ and $j$.
4. Compute the fraction of leaves in $S(\ell, j)$ that share the class $c_\ell$.

The expected value of this fraction over the sampling procedure defines the *dendrogram purity*, which equals 1 if and only if every ground-truth class forms a pure subtree.

*Implementation.* We provide an open-source implementation in our public repository. Purity is estimated by Monte-Carlo with $N = 10{,}000$ sampled pairs $(\ell, j)$. Each pair is drawn *within the same coarse Leiden cluster*; because this Leiden partition is identical across all evaluations, this sampling constraint does not introduce bias.

### B.4.2 FLAT CLUSTERING

**Normalized Mutual Information (NMI).** The uncertainty of a clustering is quantified by its *entropy*, $H(U) = -\sum_u p(u) \log p(u)$, where $p(u)$ is the probability of cluster $u$. The similarity between two clusterings $U$ and $V$ can then be measured by their *mutual information*, $I(U; V) = \sum_{u,v} p(u, v) \log \frac{p(u,v)}{p(u)p(v)}$, which captures how much knowing $V$ reduces the uncertainty of $U$. The NMI score normalizes mutual information to the range $[0, 1]$ via

$$\mathrm{NMI}(U, V) = \frac{2\, I(U; V)}{H(U) + H(V)}.$$

Because mutual information captures the overall dependency between the two label distributions, this normalization measures *global agreement* between entire clusterings rather than only local pairwise matches. Moreover, the ratio form compensates for differing cluster entropies, making the score *robust to cluster-size imbalance* and directly comparable across datasets of varying class distributions.

**Adjusted Rand Index (ARI).** To assess pairwise agreement, the *Rand index* is defined as $\mathrm{RI} = \frac{a+b}{\binom{n}{2}}$, where $a$ denotes the number of element pairs assigned to the same cluster in both $U$ and $V$, and $b$ denotes the number of pairs assigned to different clusters in both. The Rand index is corrected for chance agreement with

$$\mathrm{ARI} = \frac{\mathrm{RI} - \mathbb{E}[\mathrm{RI}]}{\max(\mathrm{RI}) - \mathbb{E}[\mathrm{RI}]},$$

placing the score in $[-1, 1]$ and emphasizing local consistency.

*Implementation.* All NMI and ARI computations use the standard implementations from `sklearn`.

### B.5 TRAINING OF BASELINES

**Louvain.** We use the Louvain implementation from the `NetworkX` library. The resolution parameter, which controls the granularity of the detected communities, is tuned by grid search in the range $[0.5, 3.0]$ on the validation graph to maximize the modularity score.

**Leiden.** For Leiden we rely on the `leidenalg` package, using the `RBConfigurationVertexPartition` objective (the standard modularity-based configuration). The resolution parameter is likewise tuned by grid search in the range $[0.5, 3.0]$ on the validation graph, and the number of refinement iterations is fixed to 10 to ensure convergence.

**Untrained GAT.** We follow exactly the same procedure as for the trained GNN experiments—using the default hyperparameters of Table 6—except that the number of training epochs is set to zero.

**Graph AutoEncoder (GAE).** We follow the non-probabilistic graph auto-encoder training procedure of Kipf & Welling (2016). The encoder is a GAT with the default hyperparameters of Table 6, while the decoder is a simple dot product. Given adjacency matrix $A$ and encoder embeddings $H$, the loss is the binary cross-entropy

$$\mathcal{L}_{\mathrm{GAE}} = -\sum_{i,j} \big[ A_{ij} \log \sigma(h_i^\top h_j) + (1 - A_{ij}) \log\big(1 - \sigma(h_i^\top h_j)\big) \big],$$

where $\sigma$ is the sigmoid function. Embeddings are trained to reconstruct $A$. We apply the same neighbor sampling, training-graph rotation, and learning-rate scheduling as in the main experiments,

monitoring performance via the validation-graph reconstruction loss. The only change in hyperparameters is a shorter training duration of 20 epochs.

**Deep Graph Infomax (DGI).** We adopt the training procedure of Veličković et al. (2018) for Deep Graph Infomax. The encoder is a GAT with the default hyperparameters of Table 6. DGI learns node embeddings by maximizing mutual information between local node representations and a global summary vector. Given node embeddings $H$ and a readout summary $s = \sigma\big(\frac{1}{n}\sum_i h_i\big)$, a corrupted graph $\tilde{G}$ is produced by randomly shuffling node features to create negative samples $\tilde{H}$. The loss is the binary cross-entropy

$$\mathcal{L}_{\text{DGI}} = -\sum_i \big[\log \sigma(h_i^\top W s) + \log\big(1 - \sigma(\tilde{h}_i^\top W s)\big)\big],$$

where $W$ is a trainable scoring matrix and $\sigma$ the sigmoid function. We use the same neighbor sampling, rotation of training graphs, and learning-rate scheduling as in the main experiments, and monitor training with the DGI objective on the validation graph. Training is limited to 20 epochs to match the GAE baseline.

## C  ADDITIONAL EXPERIMENTS

### C.1  EMBEDDING DIMENSION

A critical design choice is the number of dimensions in the embedding space produced by the GNN. If the dimensionality is too low, the model cannot adequately separate the numerous clusters. Conversely, a very high dimensionality increases algorithmic complexity and computational cost, and may even lead to dimensional collapse (Jing et al., 2021). To investigate this trade-off, we experimented with different embedding sizes, and compared their performance in Table 7. For consistency, we set the number of hidden dimensions in the GAT to $\frac{2 \times \text{Size of output embedding space}}{\text{Number of attention heads}}$.

| Embedding dimension | DP | NMI | ARI |
|---|---|---|---|
| 16 | $0.722(\pm0.004)$ | $0.755(\pm0.010)$ | $0.649(\pm0.050)$ |
| 32 | $0.745(\pm0.007)$ | $0.773(\pm0.006)$ | $0.632(\pm0.040)$ |
| 64 | $0.768(\pm0.006)$ | $0.785(\pm0.006)^*$ | $0.647(\pm0.040)$ |
| 128 | $0.783(\pm0.002)$ | $0.789(\pm0.009)^{**}$ | $0.632(\pm0.028)$ |
| 256 | $0.788(\pm0.004)^*$ | $0.773(\pm0.023)$ | $0.657(\pm0.040)^*$ |
| 512 | $0.798(\pm0.005)^{**}$ | $0.778(\pm0.009)$ | $0.700(\pm0.025)^{**}$ |

Table 7: Performance across different embedding dimensions with evaluation metrics NMI, ARI, and dendrogram purity (DP). The best score for each metric is marked with $^{**}$ and the second-best with $^*$. All metrics are computed on the test graph / heuristic clustering, and results are averaged over five runs.

Model performance generally improves as the embedding dimension increases, especially for the DP and ARI metrics. The DP score rises monotonically, indicating better hierarchical clustering quality at higher dimensions. The best NMI values occur at 64 and 128 dimensions, suggesting that an embedding size of 64 is already sufficient to capture the global cluster structure. For finer local agreement, however, higher dimensions are beneficial, as reflected by the strong ARI scores observed at 512 dimensions.

### C.2  PARAMETER OF THE SAMPLING FUNCTION

In this section, we evaluate the influence of the sampling parameter $\alpha$ on the performance of our methodology. The sampling parameter $\alpha$ controls the balance between uniform and size-proportional cluster selection in the contrastive sampling distribution. Table 8 reports the results. ARI scores are highest for small $\alpha$ (close to uniform sampling), indicating strong local cluster consistency. NMI scores peak at larger $\alpha$ (around 0.7), showing that higher values capture more global ground-truth information in the flat clustering. Dendrogram purity is maximized for intermediate $\alpha$ (around 0.4), suggesting the most coherent hierarchical structure. Overall, these trends reveal a trade-off: small $\alpha$ favors local accuracy, large $\alpha$ enhances global information, and moderate values balance the two.

| $\alpha$ | DP | NMI | ARI |
|---|---|---|---|
| 0. | $0.762(\pm0.006)$ | $0.776(\pm0.005)$ | $0.733(\pm0.005)^{**}$ |
| 0.2 | $0.776(\pm0.005)$ | $0.772(\pm0.005)$ | $0.716(\pm0.020)^{*}$ |
| 0.4 | $0.782(\pm0.004)^{**}$ | $0.773(\pm0.009)$ | $0.694(\pm0.019)$ |
| 0.6 | $0.773(\pm0.008)$ | $0.788(\pm0.007)^{**}$ | $0.630(\pm0.036)$ |
| 0.8 | $0.779(\pm0.005)^{*}$ | $0.779(\pm0.015)^{*}$ | $0.655(\pm0.028)$ |
| 1.0 | $0.770(\pm0.012)$ | $0.770(\pm0.012)$ | $0.655(\pm0.063)$ |

Table 8: Performance across different $\alpha$ with evaluation metrics NMI, ARI, and dendrogram purity (DP). The best score for each metric is marked with $^{**}$ and the second-best with $^{*}$. All metrics are computed on the test graph / heuristic clustering, and results are averaged over five runs.

## C.3 NUMBER OF NEGATIVE SAMPLES

In this section, we evaluate the impact of the number of negative examples per anchor used in the contrastive loss on the performance of our methodology. The results are reported in Table 9. The results suggest that hierarchical clustering quality, as measured by dendrogram purity, tends to improve as the number of negative anchors in the contrastive loss increases. For the flat-clustering metrics (NMI and ARI), however, no clear trend emerges, preventing any firm conclusion about their dependence on the number of negatives.

| $p$ | DP | NMI | ARI |
|---|---|---|---|
| 1 | $0.766(\pm0.004)$ | $0.774(\pm0.013)$ | $0.675(\pm0.024)^{*}$ |
| 4 | $0.769(\pm0.009)$ | $0.780(\pm0.011)$ | $0.660(\pm0.045)$ |
| 16 | $0.782(\pm0.003)^{*}$ | $0.783(\pm0.010)^{*}$ | $0.674(\pm0.048)$ |
| 32 | $0.779(\pm0.004)$ | $0.787(\pm0.012)^{**}$ | $0.641(\pm0.033)$ |
| 64 | $0.787(\pm0.004)^{**}$ | $0.774(\pm0.012)$ | $0.697(\pm0.015)^{**}$ |

Table 9: Performance across different $p$ with evaluation metrics NMI, ARI, and dendrogram purity (DP). The best score for each metric is marked with $^{**}$ and the second-best with $^{*}$. All metrics are computed on the test graph / heuristic clustering, and results are averaged over five runs.

## C.4 APPROACHING THE THEORETICAL CONDITIONS

**Cluster homophily.** Ideally, homophily would be measured by the spectral norm $\|L - L^{\circ}\|_{\mathrm{op}}$ between the graph Laplacian $L$ and the ideal block-diagonal Laplacian $L^{\circ}$, but this is infeasible for graphs with millions of nodes. As a practical alternative we use the *cut ratio*, the fraction of edges that cross between clusters: a low cut ratio indicates that most edges remain inside clusters and thus reflects strong homophily. On the validation graph the overall cut ratio is 87%; restricted to subgraphs of size 10–100 it is 77%, for size 100–1000 it is 51%, and for size 1000–5000 it is 49%. To assess the significance of these scores given the graph topology, we randomly permuted 1% of node labels and recomputed the cut ratio over 300 trials. For each case we calculated a z-score as the difference between the original score and the mean of the permuted scores, divided by their standard deviation. The corresponding p-value is the empirical probability that a random permutation yields a clustering more homophilic than the original. The resulting z-scores are -9.42 (global), -3.04 (10-100), -1.09 (100-1000), and -1.49 (1000–5000), with all p-values below 0.01, confirming that the observed homophily is highly significant for the graph topology.

**Low-pass GNN behavior.** For embeddings $H^{(\ell)}$ at layer $\ell$, the Dirichlet energy $\mathcal{E}(H^{(\ell)}) = \mathrm{Trace}\big((H^{(\ell)})^{\top} L H^{(\ell)}\big) = \sum_{i=1}^{n} \lambda_i \|H_i^{(\ell)}\|_2^2$ measures the concentration of $H^{(\ell)}$ on high-frequency eigenvectors. Normalizing by total energy gives the Rayleigh quotient $R(H^{(\ell)}) = \mathrm{Trace}\big((H^{(\ell)})^{\top} L H^{(\ell)}\big) / \mathrm{Trace}\big((H^{(\ell)})^{\top} H^{(\ell)}\big)$. A GNN acting as a low-pass filter should yield small Rayleigh quotients that decrease across layers. Across five training runs of a two-layer GAT with default parameters, the Rayleigh quotient decreases from 6.54 for the input embeddings $H^{(0)}$ to 1.29 after the first convolution $H^{(1)}$, then to 1.20 after the first activation (still $H^{(1)}$), and finally to 0.99 at the output $H^{(2)}$ on the validation subgraph. This monotonic drop confirms the expected low-pass filtering behavior.

# D  PROOFS OF SECTION 4

## D.1  PROOF OF LEMMA 1.

We begin by deriving a few spectral properties of the Laplacian $L^\circ$ of the *ideal cluster graph*, in which two nodes are adjacent if and only if they belong to the same cluster. It is well known that the Laplacian $L^\circ$ of this ideal graph has $0$ as an eigenvalue with multiplicity equal to the number of connected components—equivalently, the number of clusters (Von Luxburg, 2007). For each cluster $C_j$, the normalized indicator vector

$$u_{j,i}^\circ = \begin{cases} |C_j|^{-1/2}, & \text{if } i \in C_j, \\ 0, & \text{otherwise,} \end{cases}$$

is an eigenvector associated with the eigenvalue $0$, and these vectors form an orthonormal basis of the corresponding eigenspace.

The spectral embedding of node $i$ in the *ideal model*, using the first $k$ eigenvectors, is its coordinate vector in this basis:

$$(e_i^\circ)_j = \begin{cases} |C_j|^{-1/2}, & \text{if } i \in C_j, \\ 0, & \text{otherwise.} \end{cases}$$

Consequently, if $i, j \in C_a$ then $e_i^\circ = e_j^\circ$; and if $i \in C_a$ and $j \in C_b$ with $a \neq b$,

$$\|e_i^\circ - e_j^\circ\|_2^2 = \frac{1}{|C_a|} + \frac{1}{|C_b|} \qquad \Rightarrow \qquad \|e_i^\circ - e_j^\circ\|_2 \geq \sqrt{\frac{2}{S_{\max}}}.$$

We now view the empirical Laplacian $L$ as a perturbation of the ideal Laplacian $L^\circ$ and invoke spectral perturbation theory. Let $U_k, U_k^\circ \in \mathbb{R}^{n \times k}$ collect the eigenvectors associated with the $k$ smallest eigenvalues of $L$ and $L^\circ$, respectively. By the Davis–Kahan–type result of Yu et al. (2015), there exists an orthogonal matrix $Q \in \mathbb{R}^{k \times k}$ such that

$$\|U_k - U_k^\circ Q\|_{\mathrm{F}} \leq \frac{2\sqrt{2k}\, \|L - L^\circ\|_{\mathrm{op}}}{\lambda_{k+1}(L^\circ)},$$

where $\|\cdot\|_F$ is the Frobenius norm and $\lambda_{k+1}(L^\circ)$ denotes the $(k+1)$-th eigenvalue of $L^\circ$.

Let $e_i^s$ be the spectral embedding of node $i$ obtained from $L$. Applying the bound row-wise gives, for every node $i$,

$$\|e_i^s - e_i^\circ Q\|_2 \leq \frac{2\sqrt{2k}\, \|L - L^\circ\|_{\mathrm{op}}}{\lambda_{k+1}(L^\circ)}.$$

By the triangle inequality and the orthogonality of $Q$,

$$\|e_i^s - e_j^s\|_2 \leq \|e_i^s - e_i^\circ Q\|_2 + \|(e_i^\circ - e_j^\circ)Q\|_2 + \|e_j^\circ Q - e_j^s\|_2$$

$$\leq \frac{4\sqrt{2k}\, \|L - L^\circ\|_{\mathrm{op}}}{\lambda_{k+1}(L^\circ)} + \|e_i^\circ - e_j^\circ\|_2.$$

In particular, if $i$ and $j$ lie in the same cluster, then $e_i^\circ = e_j^\circ$ and

$$\|e_i^s - e_j^s\|_2 \leq \frac{4\sqrt{2k}\, \|L - L^\circ\|_{\mathrm{op}}}{\lambda_{k+1}(L^\circ)}.$$

A symmetric argument yields the complementary lower bound

$$\|e_i^s - e_j^s\|_2 \geq \|e_i^\circ - e_j^\circ\|_2 - \frac{4\sqrt{2k}\, \|L - L^\circ\|_{\mathrm{op}}}{\lambda_{k+1}(L^\circ)}.$$

Hence, if $i$ and $j$ belong to different clusters,

$$\|e_i^s - e_j^s\|_2 \geq \sqrt{\frac{2}{S_{\max}}} - \frac{4\sqrt{2k}\, \|L - L^\circ\|_{\mathrm{op}}}{\lambda_{k+1}(L^\circ)}.$$

Finally, for the ideal cluster graph—a disjoint union of cliques—one has $\lambda_{k+1}(L^\circ) = \frac{S_{\max}}{S_{\max}-1}$, recovering the explicit constant used earlier.

## D.2 PROOF OF THEOREM 2.

Recall that $U_k \in \mathbb{R}^{n \times k}$ is the matrix whose columns are the $k$ orthonormal eigenvectors of $L$ associated with its $k$ smallest eigenvalues. Let $U_k^\perp$ denote the matrix whose columns form an orthonormal basis of the orthogonal complement of $\mathrm{span}(U_k)$. The block matrix

$$U := \begin{bmatrix} U_k & U_k^\perp \end{bmatrix}$$

is therefore orthogonal and provides a full orthonormal basis of $\mathbb{R}^n$.

Under our structural assumption on the GNN, for input features $X \in \mathbb{R}^{n \times d}$ and weight matrix $W \in \mathbb{R}^{d \times m}$, the linearized GNN can be written

$$H = p(L)\, XW,$$

where $p$ is a polynomial filter. Using the spectral decomposition $L = UDU^\top$, this becomes

$$H = U\, p(D)\, U^\top XW.$$

This representation naturally separates the embedding into its low-frequency and residual components,

$$H = U_k\, p(D_k)\, U_k^\top XW + U_k^\perp\, p(D_k^\perp)\, (U_k^\perp)^\top XW,$$

highlighting the projection of $H$ onto the informative subspace spanned by $U_k$ and its complement along $U_k^\perp$.

Let $P_k := U_k U_k^\top$ denote the orthogonal projector onto the eigenspace spanned by $U_k$. Then

$$(I - P_k)H = U_k^\perp\, p(D_k^\perp)\, (U_k^\perp)^\top XW,$$

so the leakage of $H$ outside $\mathrm{span}(U_k)$ is controlled by

$$\begin{aligned} \|(I - P_k)H\|_{\mathrm{op}} &= \| p(D_k^\perp)\, (U_k^\perp)^\top XW \|_{\mathrm{op}} \\ &\le \|p(D_k^\perp)\|_{\mathrm{op}} \|XW\|_{\mathrm{op}}. \end{aligned}$$

Because $D_k^\perp$ is diagonal with entries given by the eigenvalues $\lambda_{k+1}, \ldots, \lambda_n$ of $L$, the operator norm of $p(D_k^\perp)$ is simply the largest absolute value of $p(\lambda_i)$ for $i > k$. Hence

$$\|(I - P_k)H\|_{\mathrm{op}} \le \left( \max_{i > k} |p(\lambda_i)| \right) \|XW\|_{\mathrm{op}} = \beta\, \|XW\|_{\mathrm{op}}.$$

Since for any matrix $A$ one has $\max_i \|A_{i,:}\|_2 \le \|A\|_{\mathrm{op}}$, it follows that for each node $i$, whose embedding is the $i$-th row $h_i$ of $H$,

$$\| h_i - (P_k H)_{i,:} \|_2 \le \|(I - P_k)H\|_{\mathrm{op}} \le \beta\, \|XW\|_{\mathrm{op}}.$$

Let $Z := U_k^\top H \in \mathbb{R}^{k \times m}$; then $P_k H = U_k Z$, so that $(P_k H)_{i,:} = (e_i^s)^\top Z$. Therefore, for any nodes $i, j \in V$,

$$\begin{aligned} \|h_i - h_j\|_2 &\le \|h_i - (P_k H)_{i,:}\|_2 + \|(e_i^s - e_j^s)^\top Z\|_2 + \|(P_k H)_{j,:} - h_j\|_2 \\ &\le 2\beta\, \|XW\|_2 + \|(e_i^s - e_j^s)^\top Z\|_2. \end{aligned}$$

Since $Z = U_k^\top H = p(D_k)\, U_k^\top XW$, we obtain

$$\|Z\|_{\mathrm{op}} \le \|p(D_k)\|_{\mathrm{op}} \|U_k^\top XW\|_{\mathrm{op}} \le \left( \max_{i \le k} |p(\lambda_i)| \right) \|XW\|_{\mathrm{op}} = \alpha\, \|XW\|_{\mathrm{op}}.$$

Consequently,

$$\|(e_i^s - e_j^s)^\top Z\|_2 \le \|e_i^s - e_j^s\|_2 \|Z\|_{\mathrm{op}} \le \alpha\, \|XW\|_{\mathrm{op}} \|e_i^s - e_j^s\|_2.$$

Combining the two displays gives the upper bound

$$\|h_i - h_j\|_2 \le \|XW\|_{\mathrm{op}} \left( 2\beta + \alpha\, \|e_i^s - e_j^s\|_2 \right).$$

A symmetric lower bound follows from the reverse triangle inequality:

$$\|h_i - h_j\|_2 \;\geq\; \|(e_i^s - e_j^s)^\top Z\|_2 - \|h_i - (P_k H)_{i,:}\|_2 - \|h_j - (P_k H)_{j,:}\|_2.$$

Using the fact that $\|(e_i^s - e_j^s)^\top Z\|_2 \geq \sigma_{\min}(Z)\,\|e_i^s - e_j^s\|_2$ and recalling that $Z = p(D_k)\,U_k^\top X W$, we obtain

$$\sigma_{\min}(Z) \;\geq\; \left(\min_{i\leq k} |p(\lambda_i)|\right)\sigma_{\min}(U_k^\top X W) = \gamma\,\sigma_{\min}(U_k^\top X W).$$

Hence,

$$\|h_i - h_j\|_2 \;\geq\; \gamma\,\sigma_{\min}(U_k^\top X W)\,\|e_i^s - e_j^s\|_2 - 2\,\beta\,\|X W\|_{\mathrm{op}}.$$

Using the bounds from Lemma 1 and substituting them into the inequalities above, we obtain the following estimates.

For nodes $i, j$ in the *same* cluster,

$$\|h_i - h_j\|_2 \;\leq\; \|X W\|_{\mathrm{op}}\left(2\,\beta + \frac{4\sqrt{2k}\,\alpha}{\lambda_{k+1}(L^\circ)}\|L - L^\circ\|_{\mathrm{op}}\right).$$

For nodes $i, j$ in *different* clusters,

$$\|h_i - h_j\|_2 \;\geq\; \gamma\,\sigma_{\min}(U_k^\top X W)\left(\sqrt{\frac{2}{S_{\max}}} - \frac{4\sqrt{2k}}{\lambda_{k+1}(L^\circ)}\|L - L^\circ\|_{\mathrm{op}}\right) - 2\,\beta\,\|X W\|_{\mathrm{op}}.$$

# E    COMPUTATIONAL COMPLEXITY ANALYSIS

We summarize here the computational costs associated with each step of our pipeline. Let $N = |V|$ denote the number of nodes, $M = |E|$ the number of edges, $d$ the embedding dimension, $k_s$ the maximum number of neighbors sampled at each GNN layer, and $k_l$ the maximum size of the Leiden pre-clusters. Table 10 reports asymptotic time and memory requirements for each stage of the method. These complexity estimates are based on the `PyTorch` implementations used for the GNN components, the `SciPy` implementation used for hierarchical agglomerative clustering, and the `leidenalg` package for the Leiden pre-clustering step.

| Step | | Time | Memory |
|---|---|---|---|
| Embeddings | Forward pass | $O(Md + Nd^2)$ | $O(Nd)$ |
| | Forward pass with sampling | $O(Nk_s^{L-1}d(k_s + d))$ | $O(Nd)$ |
| Pre-clustering | Leiden algorithm | $O(M)$ | $O(N + M)$ |
| Distance Matrix | Without pre-clustering | $O(N^2 d)$ | $O(N^2)$ |
| | With pre-clustering | $O(Nk_l d)$ | $O(Nk_l)$ |
| HAClustering | Linkage vector | $O(N^2)$ | $O(N^2)$ |
| Flat Clustering | Dendrogram cut | $O(N)$ | $O(N)$ |
| | Silhouette score | $O(N^2)$ | $O(N)$ |

Table 10: Complexity Analysis.

These results highlight the main computational bottlenecks. Embedding computation scales linearly in both $N$ and $M$, especially when neighbor sampling is applied. In contrast, operations involving pairwise distances or hierarchical clustering scale quadratically in $N$, which motivates the need for pre-clustering or highly local sampling strategies. As an illustration, Monath et al. (2021) propose a scalable HAC algorithm that mitigates these quadratic costs.

