# OpenReview forum: "Refining Heuristic-Based Bitcoin Address Clustering with Graph Neural Networks"
_ICLR.cc/2026/Conference — Submitted to ICLR 2026_

### Official Review · Reviewer_6fJg · 2025-10-25

**Soundness:** 3
**Presentation:** 3
**Contribution:** 2
**Rating:** 4
**Confidence:** 3

**Summary:**

The paper tackles Bitcoin address clustering by starting from standard heuristics that often over merge, then learning contrastive GNN embeddings designed to remain consistent with those heuristics.  Within each heuristic cluster it applies agglomerative hierarchical clustering and selects a "data-driven cut" to flag and split suspicious merges, yielding both a refined flat partition and a multi resolution view.   The authors also release a transaction graph dataset and provide theory and empirical evidence, while noting the absence of large scale ground truth user identities that would allow definitive validation.

**Strengths:**

1. Clear, practically important problem: refining over-merged Bitcoin heuristic clusters and exposing hierarchy.

2. Coherent method: contrastive GNN embeddings aligned with heuristics followed by hierarchical clustering with a data-driven cut; solid ablations and sensible diagnostics.

3. Interpretability and resources: hierarchical outputs aid analysis, and the released large transaction graph dataset increases reproducibility.

**Weaknesses:**

1. The empirical evaluation does not include recent SOTA baselines in Bitcoin clustering, so the significance of the reported improvements is unclear.

2. The supervision relies on heuristic clusters that may contain overmerges, which risks reinforcing existing errors rather than correcting them.

3. The agglomerative refinement step may not scale to very large clusters, and there is no runtime or memory analysis to assess its practicality.

4. Validation is based on intrinsic or heuristic-based metrics rather than externally verified labels, and results may be sensitive to the choice of linkage and cut threshold.

**Questions:**

1. **SOTA Baselines.** Please include head-to-head comparisons with recent SOTA pipelines for Bitcoin clustering and collapse prevention, for example Möser et al. (2022) [1], Schnoering et al. (2024) [2], Wang et al. (2024) [3] or similar recent methods. Use the same refinement pipeline and metrics for fairness.

2. **Ground truth.** Do you have externally verified labels or credible proxy labels to estimate false merges and false splits within heuristic clusters? Even a small audited subset would help calibrate the precision and recall of splits.

3. **Sensitivity.** Please report sensitivity to linkage type, distance metric, and the silhouette cut rule. Stratify by cluster size to show whether small vs large clusters require different settings.

[1] Moser, M., Narayanan, A., 2022. Resurrecting address clustering in Bitcoin. Financial Cryptography.

[2] Schnoering, H., Porthaux, P., Vazirgiannis, M., 2024. Assessing the efficacy of heuristic-based address clustering for Bitcoin. arXiv:2403.00523.

[3] Wang, X. et al., 2024. Exploring unconfirmed transactions for effective Bitcoin address clustering. The Web Conference.

> I find the paper promising, and I am open to raising my score if recent SOTA baselines and credible split-quality validation are added.

---

> ### Author Response · Authors · 2025-11-24
>
> We thank the reviewer for the detailed and constructive feedback. A new version of the paper has been uploaded. The changes are in red.

---

> > ### Author Response · Authors · 2025-11-24
> > **Weakness 2:**
> >
> > Heuristic clustering produces hard decisions with no confidence scores. A single misinterpreted transaction can merge several unrelated addresses, and the resulting cluster collapse is irreversible because the rule based procedure applies crisp binary logic without any quantitative measure of the confidence of the merge. This mechanism explains why heuristics occasionally create very large and erroneous clusters.
> >
> > Our working assumption is that heuristic clusters are generally correct. The signal to noise ratio is high because these heuristics originate from actual wallet software behavior or well known cognitive and operational patterns in Bitcoin usage. Collapses exist but remain exceptional events, typically linked to deliberate obfuscation techniques. We will make sure to take this into account and clarify it in the body of the updated paper.
> >
> > Training embeddings with a contrastive loss indeed compresses nodes inside each heuristic cluster, including those belonging to different true entities in the rare case of a collapse. However, the embeddings provide a continuous similarity score between addresses rather than a binary rule. These similarities are then used as an input to hierarchical clustering, which performs soft aggregation based on linkage scores across nodes or groups of nodes. This is fundamentally different from the hard merges produced by heuristics.
> >
> > As a result, large collapsed clusters do not appear as monolithic blocks in the dendrogram. Their internal structure becomes visible through the sequence of merge distances. Suspicious merges are characterized by unusually large linkage scores, which allows the refinement step to separate substructures that heuristics were unable to differentiate. Such a large linkage scores indicates a contradiction between the heuristic clustering (which pushed the contrastive learning to make the elements close) and the structural information signal that was treated by the GNN (which "failed" at merging the embeddings against the dynamics of the contrastive training).
> >
> > In summary, heuristics provide a reasonable initial signal but are extremely sensitive to threshold effects. Continuous embeddings and hierarchical clustering offer a more fine grained view, introduce quantitative criteria for deciding merges or splits, and therefore enable the identification and correction of collapsed clusters.

---

> > ### Author Response · Authors · 2025-11-24
> > **Weakness 3:**
> >
> > We agree that the paper currently lacks a runtime and memory analysis. We will add a discussion of computational complexity and scalability (Appendix E). Hierarchical clustering has known scalable variants, including the methods of Monath et al. [3] and Dhulipala et al. [4]. The additional experiment described in Weakness 1 also shows that a very local approach based on small transaction windows performs well in practice. For reference, all experiments were conducted on a single commercial laptop without GPU acceleration.
> >
> > [3] Monath et al. Scalable hierarchical agglomerative clustering.
> > [4] Dhulipala, et al. Terahac: Hierarchical agglomerative clustering of trillion-edge graphs.

---

> > ### Author Response · Authors · 2025-11-24
> > **Weakness 4:**
> >
> > We believe the additional experiments described in Weakness 1 provide the necessary external validation. They include evaluations with several linkage criteria (avg, complete, Ward) and three cut selection rules.

---

> > ### Author Response · Authors · 2025-11-24
> > **Question 1:**
> >
> > Regarding baseline [1], we note that the approach of Möser and Narayanan is not directly applicable to our setting. Their method restricts the analysis to a small portion of the transaction graph, namely the subset of transactions with exactly two outputs, which represents roughly thirty percent of all Bitcoin transactions. More importantly, it does not attempt to correct the common-input heuristic, even though this heuristic is the main source of cluster collapse. Obfuscation techniques such as CoinJoin specifically target this heuristic and typically involve transactions with many outputs rather than two, which places them outside the scope of [1]. That said, their work introduces a labeled dataset for these simple two-output transactions, and this dataset can indeed serve as an additional evaluation resource for our method.
> >
> > The second reference [2] will be included for certain.
> >
> > Regarding baseline [3], the proposed heuristics are promising but rely on observing the mempool in real time. Since the mempool is not recorded on the blockchain, this information is not available retrospectively. As a result, the method in [3] cannot be applied to historical data, which makes head-to-head comparison infeasible in our experimental setup.

---

> > ### Author Response · Authors · 2025-11-24
> > **Question 2:**
> >
> > The additional experiments described under Weakness 1 rely on externally verified labels or high-quality proxy labels and are designed precisely to estimate false merges and false splits in heuristic clusters. These results should directly address this question.

---

> > ### Author Response · Authors · 2025-11-24
> > **Question 3:**
> >
> > The additional experiments will report sensitivity to linkage type and cut rule. We will evaluate three linkage criteria and three cut selection rules. We are primarily focused on cosine distance, as it is directly tied to the contrastive loss used during training. We are also considering how to refine the treatment of clusters based on their size, and we look to include an analysis in this direction.

---

> ### Author Response · Authors · 2025-11-24
> **Weakness 1:**
>
> We acknowledge that evaluating improvements without large scale ground truth is challenging. To address this, we propose two additional experiments.
>
> - First, using the dataset of Schnoering and Vazirgiannis [1], which provides an address-to-entity mapping for roughly one hundred thousand Bitcoin addresses, we extracted all transactions involving at least two labeled addresses. We sampled five hundred such transactions and constructed local graphs using the same procedure as in the paper, except that each graph is centered on the labeled transaction rather than a coinbase transaction. These graphs contain only a few hundred transactions each, and the labels allow us to evaluate clustering quality on labeled address pairs. A correct prediction is defined as grouping two addresses when they belong to the same entity and separating them otherwise. We compared heuristic clustering, a purely GNN-embedding-based clustering that ignores heuristics, and a hybrid approach where GNN embeddings refine the heuristic clusters. The GNNs used here are not retrained but directly reused from the contrastive training stage. The hybrid method yields the best performance, with gains of approximately 10\% in accuracy and F1, and a reduction of false positives by roughly 50\%. These results are computed on more than seven hundred labeled address pairs.
>
> - Second, we used the CoinJoin-labeled dataset introduced in [2] to evaluate how well our refinement resists adversarial patterns specifically designed to fool heuristics.
>
> We will also include all newly constructed labeled graphs in the dataset so that the evaluation can be fully reproduced.
>
>
> [1] Schnoering and Vazirgiannis. Bitcoin research with a transaction graph dataset.
> [2] Schnoering and Vazirgiannis. Heuristics for detecting coinjoin transactions on the bitcoin blockchain.

---

> ### Comment · Reviewer_6fJg · 2025-11-26
> **Thank you**
>
> Thank you for the detailed rebuttal and the additional experiments with entity labels, CoinJoin data, sensitivity analysis, and scalability discussion, which address several of my earlier concerns. That said, I still find the SOTA comparison insufficient for an ICLR paper: the new results (Table 2) mainly compare heuristics, GNN only, and your hybrid method on locally constructed graphs using datasets from prior work, but they do not implement or adapt full competing refinement pipelines (e.g., Moser–Narayanan style or other recent refinement methods) end to end on the same task and metrics. In my view, this remains closer to an internal ablation than a true head to head SOTA benchmark, and without such a comparison it is hard to evaluate the practical significance of the gains. Overall, I find the paper promising and relevant, but I think it needs stronger SOTA baselines to meet the ICLR bar, so I will keep my score unchanged.

---

### Official Review · Reviewer_A8uE · 2025-10-31

**Soundness:** 2
**Presentation:** 2
**Contribution:** 2
**Rating:** 2
**Confidence:** 3

**Summary:**

This paper presents a two-stage methodology for refining heuristic-based Bitcoin address clusters, which are known to be flat and prone to erroneous merges. The work proposes first training a GNN using a contrastive loss to learn embeddings that are consistent with the initial heuristic clusters. In the second stage, agglomerative hierarchical clustering is applied to these embeddings within each heuristic cluster. By cutting the resulting dendrogram at a learned threshold, the method aims to identify and split these suspicious merges.

**Strengths:**

S1. The paper addresses a significant limitation of existing Bitcoin analysis methods.

S2. The release of a new, and publicly available dataset of Bitcoin transaction graphs is a contribution.

**Weaknesses:**

W1. The stated goal is to correct flawed heuristic clusters, specifically "cluster collapse”. However, the GNN is trained with a contrastive loss where positive pairs are drawn from the same heuristic cluster, and negative pairs from different clusters. This training objective is in direct opposition to the goal of finding and separating distinct user entities that were erroneously merged within that same cluster C_i. The paper fails to provide a convincing justification for why an embedding space trained to compress a cluster should be suitable for de-agglomerating it.

W2. The theoretical analysis in Section 4 appears disconnected from the paper's refinement objective. The theory provides no theoretical justification for the refinement step.

W3. The current baselines are generic clustering or representation learning methods. A proper comparison would be against other methods designed to solve the same problem: refining heuristics.

W4. It is recommended authors open-source the code to reproduce the results.

**Questions:**

Q1. For the Training Objective, could please elaborate on the core intuition of the methodology? The GNN is trained with a contrastive loss that explicitly pulls all nodes in a heuristic cluster C_i together. How does this training objective produce an embedding space that is suitable for the second stage, which aims to find and separate erroneously merged nodes within that same cluster C_i?

---

> ### Author Response · Authors · 2025-11-24
>
> We thank the reviewer for the helpful feedback. Our responses are provided below. A new version of the paper has also been uploaded. the changes are in red.

---

> > ### Author Response · Authors · 2025-11-24
> > **Weakness 1 / Question 1**
> >
> > Heuristic clustering produces hard decisions with no confidence scores. A single misinterpreted transaction can merge several unrelated addresses, and the resulting cluster collapse is irreversible because the rule based procedure applies crisp binary logic without any quantitative measure of the confidence of the merge. This mechanism explains why heuristics occasionally create very large and erroneous clusters.
> >
> > Our working assumption is that heuristic clusters are generally correct. The signal to noise ratio is high because these heuristics originate from actual wallet software behavior or well known cognitive and operational patterns in Bitcoin usage. Collapses exist but remain exceptional events, typically linked to deliberate obfuscation techniques. We will make sure to take this into account and clarify it in the body of the updated paper.
> >
> > Training embeddings with a contrastive loss indeed compresses nodes inside each heuristic cluster, including those belonging to different true entities in the rare case of a collapse. However, the embeddings provide a continuous similarity score between addresses rather than a binary rule. These similarities are then used as an input to hierarchical clustering, which performs soft aggregation based on linkage scores across nodes or groups of nodes. This is fundamentally different from the hard merges produced by heuristics.
> >
> > As a result, large collapsed clusters do not appear as monolithic blocks in the dendrogram. Their internal structure becomes visible through the sequence of merge distances. Suspicious merges are characterized by unusually large linkage scores, which allows the refinement step to separate substructures that heuristics were unable to differentiate. Such a large linkage scores indicates a contradiction between the heuristic clustering (which pushed the contrastive learning to make the elements close) and the structural information signal that was treated by the GNN (which "failed" at merging the embeddings against the dynamics of the contrastive training).
> >
> > In summary, heuristics provide a reasonable initial signal but are extremely sensitive to threshold effects. Continuous embeddings and hierarchical clustering offer a more fine grained view, introduce quantitative criteria for deciding merges or splits, and therefore enable the identification and correction of collapsed clusters.

---

> > ### Author Response · Authors · 2025-11-24
> > **Weakness 2**
> >
> > The theoretical analysis clarifies why the embedding stage is meaningful for the refinement objective. It shows that if the graph is sufficiently close to an ideal cluster graph, then embeddings produced by a GNN admit a dendrogram with a threshold that perfectly recovers the ground truth. Without such guarantees, it would be unclear whether learning embeddings is relevant to a clustering task.
> >
> > We acknowledge that the theory does not address the selection of the optimal cut threshold in the refinement step. This is an interesting direction for future work, and we agree that a more explicit theoretical treatment of the refinement criterion would strengthen the overall framework. In particular, the threshold selection could be based on the bounds proposed in the proof. Nonetheless, the current definition of the bounds requires untractable spectrum computation, leading to us using silhouette score as a loose surrogate. Finding an efficient way of computing that bound could be an interesting future contribution but is outside the scope of the current method.

---

> > ### Author Response · Authors · 2025-11-24
> > **Weakness 3**
> >
> > To our knowledge, only two existing works explicitly address the refinement of heuristic based Bitcoin clustering, as mentioned in the introduction. We are currently running experiments on labeled datasets. If feasible, we will integrate these baselines.

---

> > ### Author Response · Authors · 2025-11-24
> > **Weakness 4**
> >
> > The full codebase will be open sourced and fully reproducible using the accompanying dataset. An initial version is already provided with this submission, and is accessible as supplementary material on OpenReview. The final repository will include the complete training and evaluation pipeline.

---

### Official Review · Reviewer_RZgt · 2025-11-05

**Soundness:** 2
**Presentation:** 3
**Contribution:** 1
**Rating:** 2
**Confidence:** 4

**Summary:**

The paper proposes to use graph neural networks with slightly modified loss function to compute embeddings in Bitcoin graphs to address clustering.

**Strengths:**

Evaluation of different Graph neural network approaches on address clustering in bitcoin.

The code and the data will be open sourced.

**Weaknesses:**

The proposed approach appears fairly straightforward, with limited novelty in the design of the loss function. Address clustering is performed by generating GNN-based embeddings, followed by the application of a standard clustering algorithm.

Also, the method does not seem scalable to real-world Bitcoin network data, where graph sizes are extremely large. This scalability limitation is likely one of the key reasons why GNN-based methods are rarely applied directly in practice for such blockchain analysis tasks anyway.

The theoretical results seem rehash of existing theorems.

**Questions:**

None.

---

> ### Author Response · Authors · 2025-11-24
>
> We thank the reviewer for the helpful comments. Below are our responses to each point. A new version of the paper has also been uploaded. the changes are in red.

---

> > ### Author Response · Authors · 2025-11-24
> > **Weakness 1**
> >
> > You say: "The proposed approach appears fairly straightforward, with limited novelty in the design of the loss function. Address clustering is performed by generating GNN-based embeddings, followed by the application of a standard clustering algorithm."
> >
> > While the pipeline can indeed be summarized simply as learning embeddings followed by clustering, the contribution goes beyond this high level description. The contrastive loss is specifically tailored to heuristic clusters. Moreover, the subsequent clustering induces a hierarchy in the embedding space, which yields a quantitative criterion for identifying inconsistencies inside heuristic generated clusters.
> >
> > Although hierarchical clustering is not novel, using it as a refinement layer supported by theoretical guaranties and empirical validation provides a multi resolution view of user communities that standard flat clustering cannot capture. This two stage strategy, consisting of contrastive embeddings and hierarchical refinement, is also applicable to any node level clustering task on graphs.
> >
> > For these reasons, we view the overall approach as a coherent and original methodological contribution.

---

> > ### Author Response · Authors · 2025-11-24
> > **Weakness 3**
> >
> > You say "The theoretical results seem rehash of existing theorems.".
> >
> > The theoretical section does not aim to introduce new mathematical theorems but rather to adapt existing perturbation and spectral results to the specific context of GNN based embeddings used for refining heuristic clusters. To our knowledge, no previous work provides conditions under which GNN or contrastive embeddings admit a perfect hierarchical cut that recovers heuristic clustering.
> >
> > Since the embeddings constitute the core of the method, providing guarantees on their separability is essential. Classical tools, such as Davis–Kahan type bounds, are therefore specialized to this setting. Their application to the geometry of contrastive and GNN derived embeddings, and to the structure of the induced dendrograms, is new in this context and offers a principled justification for our refinement procedure.

---

> ### Author Response · Authors · 2025-11-24
> **Weakness 2**
>
> You say: "Also, the method does not seem scalable to real-world Bitcoin network data, where graph sizes are extremely large"
>
> The Bitcoin graph is indeed extremely large, but several aspects of our methodology address scalability constraints.
>
>
> 1) **Scalability of GNN embedding construction.** Message passing GNNs (such as the ones we use in this approach) are very scalable on sparse graphs. Indeed, their complexity is the product of the number of nodes (since each node is updated) and the number of messages each node receives (i.e. at most its number of neighbors). The two points below show how we control that complexity.
> 2) **Controlling the number of nodes.** *Time-locality of the approach.*
> Training is performed on limited temporal windows: only addresses used in that interval are used, limiting the number of nodes of the sampled subgraph. This ensures that the training subgraph has temporal coherence and looks close to real-life applications, where we would work with the n most recent blocks and not need to consider addresses that have not been used recently. Models trained on a given interval generalize well to intervals more than a year apart, suggesting that this subgraph-sampling approach is valid.
> 3) **Controlling the number of messages:** *Neighbor sampling.*
> The embedding phase relies on neighbor-sampling, a standard and well validated technique for handling large degree nodes and large scale message passing. In practice, this stage scales efficiently, as it limits the number of neighbors, therefore of messages. Once again, the good generalization shows that this approximation does not sacrifice excessive signal.
> 4) **Scalability of hierarchical clustering.**
> While naïve HAC is quadratic, scalable variants exist, including the methods of Monath et al. [1] and the TeraHAC framework for trillion edge graphs [2]. In our setting, clusters tend to be well connected, so the distance matrices are close to block diagonal, which significantly lowers practical computational cost. Refinement can also operate on subgraphs or around a specific entity when needed. We propose to add this in the paper as an additional support for our method.
> 5) **Preliminary results on labeled data.** (Section 6.3)
> Experiments on labeled subsets show strong performance even with subgraphs containing only a few hundred transactions (see the answer to review 6fJg), which further supports the idea that local training is sufficient.
>
> Overall, the combination of local training, sampling based GNNs, scalable HAC techniques, and the structural properties of Bitcoin clusters makes the approach compatible with real world blockchain scale analysis.
>
> [1] Monath, Nicholas, et al. Scalable hierarchical agglomerative clustering.
> [2] Dhulipala, Laxman, et al. "Terahac: Hierarchical agglomerative clustering of trillion-edge graphs."

---

### Meta-Review · Area_Chair_rkyM · 2026-01-05

**Summary:**

This paper studies Bitcoin address clustering. Unlike existing methods that produce flat cluster assignments, the authors propose refining heuristic-obtained clusters by grounding the clustering process in contrastive embeddings. However, the submission has several significant shortcomings. It fails to provide comparisons with state-of-the-art baselines, the theoretical analysis is not well aligned with the paper’s refinement objective, and the proposed approach does not appear to scale to real-world, large-scale Bitcoin datasets. Therefore, I recommend rejecting this paper.

**Reviewer Concerns:**

The rebuttal partially addresses the reviewers’ concerns. Several key issues remain outstanding. The rebuttal does not include comparisons with state-of-the-art baselines, the theoretical analysis remains insufficiently connected to the paper’s stated refinement objective, and concerns regarding scalability to real-world, large-scale Bitcoin datasets are not adequately addressed. As such, the core weaknesses identified in the initial reviews persist.

**Reviewer Scores:**

Based on the rebuttal and discussion, I do not believe the reviewers would have substantially changed their scores. While some clarifications were provided, the key concerns raised by each reviewer remain unresolved. As a result, the overall assessment of the paper would likely remain unchanged.

---

### Decision · Program_Chairs · 2026-01-26

Reject